# Multi-Modal Data Correspondence for the 4D Analysis of the Spine with Adolescent Idiopathic Scoliosis

**DOI:** 10.3390/bioengineering10070874

**Published:** 2023-07-24

**Authors:** Nicolas Comte, Sergi Pujades, Aurélien Courvoisier, Olivier Daniel, Jean-Sébastien Franco, François Faure, Edmond Boyer

**Affiliations:** 1Anatoscope, 38330 Montbonnot-Saint-Martin, France; nicolas.comte@inria.fr (N.C.); francois.faure@anatoscope.com (F.F.); 2Inria, Université Grenoble Alpes, CNRS, Grenoble INP, LJK, 38000 Grenoble, France; sergi.pujades-rocamora@inria.fr (S.P.); jean-sebastien.franco@inria.fr (J.-S.F.); edmond.boyer@inria.fr (E.B.); 3TIMC-IMAG, University Grenoble Alpes, CNRS, UMR 5525, VetAgro Sup, Grenoble INP, CHU Grenoble Alpes, 38000 Grenoble, France; odaniel@chu-grenoble.fr; 4Grenoble Alps Scoliosis and Spine Center, Grenoble Alps University Hospital, Bvd de la Chantourne, CEDEX 09, 38043 Grenoble, France

**Keywords:** motion capture, idiopathic scoliosis, kinematics, spine, subject-specific modeling

## Abstract

Adolescent idiopathic scoliosis is a three-dimensional spinal deformity that evolves during adolescence. Combined with static 3D X-ray acquisitions, novel approaches using motion capture allow for the analysis of the patient dynamics. However, as of today, they cannot provide an internal analysis of the spine in motion. In this study, we investigated the use of personalized kinematic avatars, created with observations of the outer (skin) and internal shape (3D spine) to infer the actual anatomic dynamics of the spine when driven by motion capture markers. Towards that end, we propose an approach to create a subject-specific digital twin from multi-modal data, namely, a surface scan of the back of the patient and a reconstruction of the 3D spine (EOS). We use radio-opaque markers to register the inner and outer observations. With respect to the previous work, our method does not rely on a precise palpation for the placement of the markers. We present the preliminary results on two cases, for which we acquired a second biplanar X-ray in a bending position. Our model can infer the spine motion from mocap markers with an accuracy below 1 cm on each anatomical axis and near 5 degrees in orientations.

## 1. Introduction

Adolescent idiopathic scoliosis (AIS) is a 3D spinal disorder affecting 1–3% of the population and that evolves during the period of growth [1]. There is currently a lack of knowledge about its etiopathogenesis. Several factors are investigated, such as genetics, hormones or biomechanical disorders [2]. Scoliosis is usually diagnosed and monitored from 2D X-rays [3], but the recent literature shows that the new static 3D characterization methods, such as the severity-index [4] computed from different 3D descriptors of the deformities (torsion index, Cobb angle, axial rotations, …), provide valuable insights into the progression of scoliosis. Despite the advances in the static characterization, the dynamic behavior of the condition remains poorly studied in the scientific literature. However, motion capture technology provides a promising approach to analyze the three-dimensional movement of patients with scoliosis.

Motion capture provides a superficial analysis of the patient’s motion but does not directly capture the actual dynamics of the spine inside the body. Several works [5,6,7] have investigated how to describe the spinal alignments in motion by acquiring 3D trajectories with mocap markers positioned on the palpable spinous process of the vertebrae. These methods, however, are highly sensitive to the palpation task and underestimate the coronal curvatures measurements [8] without numerical correction regarding the true anatomy [6]. In addition, these methods cannot provide a detailed analysis in rotations at each vertebra level.

A complementary approach is the use of a kinematic model in charge of the reconstruction of the vertebra dynamics according to external constraints. Rigid-body modeling is the usual approach, in which the bones are defined as a set of rigid-bodies articulated with joints that allow different degrees-of-freedom (DOFs) in rotation and/or translation. Rigid-body models have already been used in the scope of static treatment correction prediction and analysis [9,10,11]. Overbergh et al. [7] proposed and validated a workflow allowing the reconstruction of a subject-specific model that can be used towards marker-based motion capture analysis. They leveraged the recent advances in low-dose biplanar radiography provided by the EOS Imaging System (EOS Imaging, Paris, France) to validate their kinematic predictions. However, their method was evaluated for adult spinal deformities and requires a sensitive and time-consuming palpation task which cannot be easily implemented on young AIS patients.

The aim of this study is to build a subject-specific kinematic model of patients with AIS that captures both their internal and external specificities. The digital twin can then be driven with mocap markers on the back of the patient, which yields the actual kinematics of the spine. We present a preliminary validation of the predictions against 3D spine reconstructions of two patients who performed a lateral bending.

## 2. Materials and Methods

### 2.1. Collected Data

Our dataset consists of 8 patients aged between 8 and 16 years with AIS (Cobb angle range: 14–68${}^{{}^{\circ}}$) and without any treatment history. They were included in our study following the IRB CPP Ile de France 2 on 20 July 2020: $n^{{}^{\circ}}$ ID RCB: 2020-A01071-38. All parents and patients received an information letter. Two data modalities were collected in our institution for each subject:A biplanar X-ray of their trunk made with an EOS imaging systemA surface scan of the back using an Occipital Structure Sensor Mark II (XRPro, LLC, Saratov)

During all acquisitions, the patients wore a set of 18 radio-opaque markers positioned in the neighborhood of 5 vertebrae: T1, T4, T7, T10 and L03. The marker placement does not require a precise palpation. The markers can be identified and located in the X-ray images and the surface skin mesh (Figure 1 and Figure 2).

To validate the kinematic predictions of the model, X-rays of two voluntary patients, with Cobb angles of 14${}^{{}^{\circ}}$ and 29${}^{{}^{\circ}}$, were captured in left and right lateral bending. These images will be used to quantify the inference of the spine motion inside the body in different poses.

### 2.2. Data Processing

Some manual steps are needed to annotate the images. The first step is to make the semi-automated reconstruction of the 3D geometries of the spine from the biplanar radiographs with the SterEOS software 1.8.99.21R (copyright 2015 EOS Imaging) [12] (EOS Imaging, Paris, France). Then, the radio-opaque markers, visible in the images, are located in 2D and their 3D position is computed using the calibration information available in the DICOMs metadata [13].

From the surface skin mesh, the marker locations are identified. As the surface scan and the EOS reconstruction are defined in a different global frame, we use a rigid registration to bring them in the same frame. Namely, we compute the translation and rotation that minimizes the distance between the 3D markers identified in both modalities. Let us note that as the pose might be slightly different in both acquisitions, we use RANSAC [14] to filter out markers whose positions might have significantly changed with the patients pose. This registration provides a first association between the surface scan and the spine 3D reconstruction.

### 2.3. The Subject-Specific Kinematic Model

To create a 4D numerical twin of the patient, we leverage the Anatoscope technology based on “Anatomy transfer” [10,15,16]. This method deforms an initial anatomic model to capture the internal and external shapes of the patient with rigid and elastic registration processes. The resulting avatar can then be used for biomedical simulations, namely, the parameters of the model can be optimized so that the skin of the model matches the mocap markers while enforcing biomechanical constraints on the spine behavior.

Our kinematic model has $N_{x}=18$ articulated rigid-bodies $x_{i}$ defined in positions $p_{i}\in R^{3}$ and rotations $R_{i}\in R^{3\times 3}$, corresponding to each *i* thoracolumbar vertebra and the pelvis. In addition each bone is associated with a set of shape parameters s that modify the model geometries. The bones are connected by K=17 joints of 6 degrees-of-freedoms (DOFs) in translations and rotations, as defined by Ignasiak et al., 2016 [17]. We followed the modification applied by Koutras et al., 2021 [18], allowing for a symmetrical definition of the vertebral motion (Table 1). The position of the joints is defined in the middle of the segment drawn between the two adjacent endplate centroids. Their orientation is defined accordingly to the inferior rigid body. The model also has a skin surface that is rigged by the articulated rigid bodies.

The first step of the registration process is to change the pose x and the shape s parameters of the model so that the models’ spine fits the 3D reconstructed spine and the models’ skin surface captures the surface scan. As the shape parameters s only capture several deformation models, the obtained geometries do not precisely match the patient-specific geometries. Thus, in a second step, we refine the geometries of the model’s vertebrae and skin to match the patient’s observations (EOS geometries and surface scan). Once the patient surfaces are captured by the model, the shape parameters are fixed. Let us note that the patients pose during the surface scan is slightly different than the pose in the radiography, thus, the location of the markers $m^{A}$ on the resulting model differs from the marker locations $m^{D}$ on the X-rays. To fix this issue, we transfer the $N_{m}$ markers positions $m^{S}\in R^{3\times N_{m}}$ located on the surface scan onto the model skin mesh. We identify the closest mesh face of the model to a marker and define the marker location on the model mesh using the barycentric coordinates of the face vertices. Then, we use a temporary set of pose parameters $x^{\prime }$ that are optimized so that the model markers $m^{A}\left(x^{\prime }\right)$ match the ones in the X-rays $m^{D}$. This effectively changes the model skin surface to match the pose of the patient in the EOS device.

The marker-based optimization is computed as
(1)$$\hat{x^{\prime }}=\underset{x^{\prime }}{\arg\;\min}\;\left(\sum_{i=1}^{i=N_{m}}\left|\right|m_{i}^{A}\left(x^{\prime }\right)-m_{i}^{D}{\left|\right|}^{2}+E_{model}\left(x^{\prime }\right)\right),$$
where the energy $E_{model}$ is a regularization term enforcing anatomic constraints on the joints of the model.

The resulting model skin surface matches the pose of the back surface during the X-ray acquisitions. Thus, we create a synced skin and spine model by disregarding the temporal parameters $x^{\prime }$ and associating the current optimized skin to the original model parameters x obtained during the first registration process. As a result, the anatomical model is a numerical twin of the patient, including the skeleton and the skin rigged with common model parameters. The association of the skin and spine is performed on the pose observed during the X-ray acquisitions with the help of the radio-opaque markers. Given a new set of markers, the model parameters can be optimized using Equation (Equation 1) and obtain the skin model matching the input markers, as well as a prediction of the spine geometry inside the body.

### 2.4. Accuracy of the Anatomical Model

To evaluate the quality of our model, we compute several metrics related to the accuracy in shape, positions and orientations of the vertebrae. The ground-truth measurements of the vertebrae are based on the 3D annotations provided by the SterEOS software. We compare our model meshes with the reconstructed ones.

The vertebrae location is defined with the center of mass of the vertebra mesh. The euclidean distance between the corresponding ground truth and model meshes is then computed with their mean absolute difference on each anatomical axis. The differences in orientations are given by the 2D projection of a vertebra orientation vector, computed according to the recommendations of the International Society of Biomechanics (ISB) [19] on a given anatomical plane. The resulting angle between the SterEOS measurement and our model is then measured for each vertebra on each plane. To assess the quality of the model vertebrae geometry, we compute the absolute mean and standard deviation of the point-to-surface distances between the model geometries and the EOS 3D reconstructed spines.

As the body surface of the back during the radiograph acquisitions is not available, we evaluate the model fit to radiographs by computing the 3D euclidean distances between the radio-opaque markers on the model and in the X-ray. We also quantify the contribution of the model skin correction step used to reconstruct the pose of the back during the X-ray acquisition.

### 2.5. Validation of the Kinematic Predictions

We validate the motion of the spine inside the body predicted by our model as follows. X-rays from two voluntary subjects with AIS were acquired in different poses in the EOS imaging system. The standard pose, standing with hands on the cheeks, was used to create the digital twin (Figure 2a). Then, two other poses, right and left lateral bending, were also acquired (Figure 2b). From the resulting images, the marker positions were identified and located and the 3D spine model reconstructed. The markers of these poses are used to drive the model, and the obtained 3D spine is compared to the reconstructed one. Specifically, given a set of 3D markers in the X-rays $m^{D}$, the model parameters x can be optimized so that the model markers $m^{A}$ best match the input markers $m^{D}$ by optimizing Equation (Equation 1).

Let us note that the 3D reconstruction of the vertebrae in lateral bending is not straightforward due to the overlapping of the different bones on the profile X-ray (Figure 2b right). Thus, the vertebra details needed for the reconstruction are difficult to extract: the computed geometries of the vertebrae do not accurately match the image and do not match the geometries obtained in the standing pose. To overcome this issue, we rigidly registered the model vertebra geometries obtained in the standing position to the reconstructions in bending position. This step computes the optimal rigid transformation of each vertebra to minimize the projected distances of the model geometries in standing and those obtained in bending.

With this procedure, we obtained the 3D rigid location and orientation of each vertebra from the bending images. Thus, we compute the orientation errors with respect to the predictions by using the intrinsic Euler angles defined by the ISB XYZ sequence (coronal, axial, sagittal). The accuracy in position is given as described in Section 2.4.

## 3. Results

We evaluated the created digital twin in two settings. We first quantified the capability of the model to capture the data in the standing position, and then we evaluated the precision of the model in the bending position. For each position, we assess the external accuracy, i.e., how well does the model fit the 3D markers, as well as the internal accuracy, i.e., how well does the model capture the shape and position of the 3D spine inside the body?

### 3.1. Accuracy of the Subject-Specific Model in Standing

#### 3.1.1. External Accuracy

We tested our method to correct the pose of the model skin according to the 3D positions of the markers in the X-rays. In Figure 3, it appears that we were able to have a significant gain in accuracy in the surface reconstruction of the model, reflected by the position of the markers, with the correction step. The average distance error decreases from 9.86 mm (std: 8.10 mm) to 4.47 mm (std: 2.70 mm).

#### 3.1.2. Internal Accuracy

The measurements in positions are made by computing the center of mass of each mesh. The orientations of each thoracic and lumbar vertebra (T01-L05) are produced according to the ISB recommendations. The shape accuracy is given for each vertebra by computing the mean of the absolute point-to-surface distances. The results are detailed for each vertebra on Table 2. The error in positioning T01 is due to a model registration error on a unique patient whose vertebra has a particular shape. Despite this result, the model performs well in capturing the morphological specificities of the patient’s spine.

### 3.2. Accuracy of the Subject-Specific Model in Bending

From the X-ray images, the radio-opaque marker positions were manually extracted and their 3D location triangulated. Their positions serve as inputs for the inverse-kinematics problem.

For this experiment, we removed the two most lateral markers at the T04 level, as these markers are subject to the movement of the scapula, which is not included in our model. We can notice that their positions have not correctly fit their target according to the anatomical model for these two subjects with an average 3D error of 6.68 mm (std: 1.25 mm).

Let us note that for one subject, the right bending pose resulted in most of the markers being out of the X-ray frontal plane view. Thus, we do not report the metrics on this case.

#### 3.2.1. External Accuracy

After the marker optimization, the model markers reached their corresponding targets with an average distance error of 4.66 mm (std: 2.14 mm).

#### 3.2.2. Internal Accuracy

We evaluated the accuracy of our predictions in positions by comparing the center of mass for each vertebra. The errors in rotations are given by comparing the intrinsic Euler angles between the ground-truth and predictions according to the XYZ order given by the ISB recommendations [19]. The results are presented for each vertebra in Table 3.

The 3D accuracy of the predicted positions is close to the actual cm on average with 1.07 cm (std: 0.42 cm). It can be noted that the predictions are affected by a global lateral shift in the side of the movement (Figure 4) highlighted on the mediolateral axis (MAE 8.49 mm, std: 5.13 mm). Despite this error, the orientation in the corresponding plane (coronal) is closer to the expectations (4.57${}^{{}^{\circ}}$, std: 4.53${}^{{}^{\circ}}$, Table 3). The greater error in the L05 transverse rotation can be due to a lack in superficial constraints (i.e., markers) in this region.

## 4. Discussion

In this study, we presented a semi-automatic workflow that allows the creation of a 4D numerical avatar of patients with AIS that reflects their inner and external anatomy. The inputs were captured using safe and low-dose imaging methods and do not need the sensitive and time-consuming task of palpation that cannot be applied on young subjects in daily clinical usage. An anatomical avatar is deformed in order to capture the internal (spine) and external (skin) specificities of the patient obtained from the different modalities. The model was able to capture the vertebra geometries with a mean error below the mm (0.29 mm, std: 0.28 mm), allowing us to compute several descriptors of the vertebra positions and orientations automatically. One major challenge is to recover the external shape of the patient during the X-ray acquisitions, as the patient’s pose is necessarily different from the surface scan acquired separately. A solution can be found in the introduction of 3D sensors during the X-ray acquisition [20]. We proposed a method that leverages the radio-opaque markers, visible in both the X-ray images and surface scan, to correct the pose of the model’s back. This additional step in our workflow allows us to capture the change in pose of the patient in standing and to increase the correspondence with the markers in the X-rays from 9.86 mm (8.10 mm) on average to 4.47 mm (std: 2.70 mm). However, we notice the difficulty for our model to fit the lateral markers, particularly on the upper part of the body. This can be explained by the fact that we are not modeling the shoulder girdle, and some markers, positioned near T04, for instance, are placed on the scapula.

In a preliminary study, we validate the predictions of the kinematic model with secondary X-rays of two voluntary patients, who were asked to make lateral bendings (left and right, Figure 2). One capture was rejected due to the low visibility of the markers. The marker positions were used as inputs in the simulator, and the predicted positions and orientations of the vertebrae were compared to the 3D X-ray reconstructions.

The creation of a thoracolumbar kinematic model of patients with the scoliosis condition was also investigated by Overbergh et al., 2020 [7]. They were able to make a subject-specific model of their patients comprising bone geometries and a set of superficial markers. Then, they compared the kinematic predictions of the models against secondary X-rays of the subjects in different poses. However, this workflow was designed towards adult spinal deformity analysis and did not integrate the external surface of the back. We proposed a method that leverages the 3D acquisition of the back to avoid the precise and time-consuming task of palpation. Thus, this protocol can be used by less experienced medical staff.

Our predictions are in the range of values of those reported in [7] in positions and orientations, except for the mediolateral axis measurements. In this case, the predicted spine was affected by a global lateral shift on the side of the bending. The orientations of the vertebrae were in the range of acceptable values with an average accuracy below $6^{{}^{\circ}}$ on the intrinsic Euler rotations. The reconstructions in the lumbar part of the spine showed a more important error, particularly on the axial rotations. This is also the most flexible part of the spine where the rotations can be underestimated by our kinematic model.

Some limitations can be noted such as the joint stiffness that is defined using the Ignasiak et al., 2016 [17] and Koutras et al., 2021 [18] values obtained from healthy adult observations. The optimization processes can be planned in order to estimate more specific kinematic parameters such as the joint stiffness or the marker constraint. The addition of radio-opaque markers at the hips would allow us to improve the model predictions in the lower part of the spine and to validate predictions with the pelvis. Furthermore, the recruitment of more patients, with and without AIS, would provide a more comprehensive overview of the model performances.

## 5. Conclusions

Combined with 3D X-ray imaging, the characterization of the spine in motion would provide valuable insights about the scoliosis condition of the patient. This analysis is challenging since there is no non-invasive method to capture the vertebral motion in vivo in the young patient. In this study we investigated the use of a subject-specific kinematic model obtained from low-dose biplanar X-rays, a surface scan and a set of radio-opaque markers. As a preliminary result, we evaluated the kinematic behavior of the model against secondary biplanar X-rays of two patients in lateral bending. We show that the kinematic predictions are close to the radiograph observations with an accuracy near 1 cm in 3D position and 5${}^{{}^{\circ}}$ in orientation.

Future work can be considered, such as the addition of markers or the optimization of the physical model constraints. Leveraging bigger cohorts of patients in motion will further allow us to better characterize the individual physical properties of the patients. Ultimately, a comprehensive understanding of scoliosis using 4D avatars will lead to improved scoliosis classification, its diagnosis and treatment with biomedical simulation.

## Figures and Tables

**Figure 1 bioengineering-10-00874-f001:**
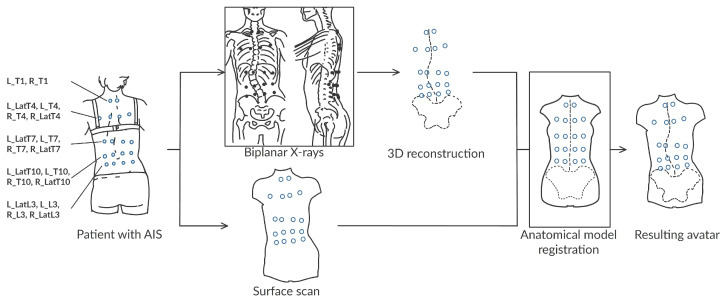
Summary of our workflow with the annotated data. A set of radio-opaque markers (blue circles) are placed on the back approximately according to different vertebra levels. No palpation is required. The acquisition of their 3D location in both modalities (surface scan and biplanar X-rays) allows the spatial correspondence between the internal and external structures (skin and spine).

**Figure 2 bioengineering-10-00874-f002:**
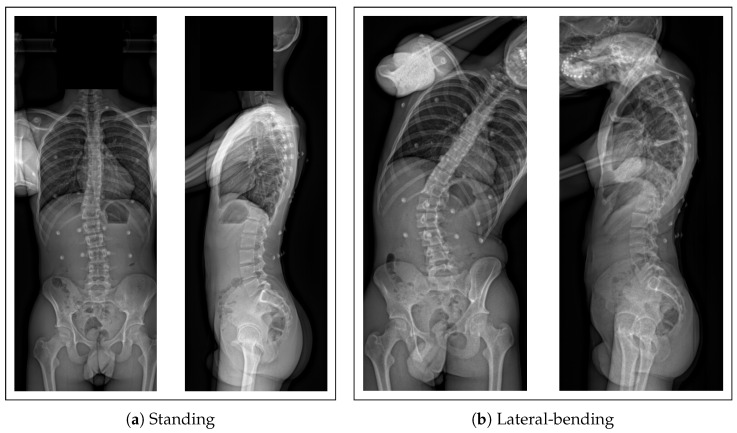
Biplanar X-rays of the same patient in standing (**a**) and lateral bending (**b**).

**Figure 3 bioengineering-10-00874-f003:**
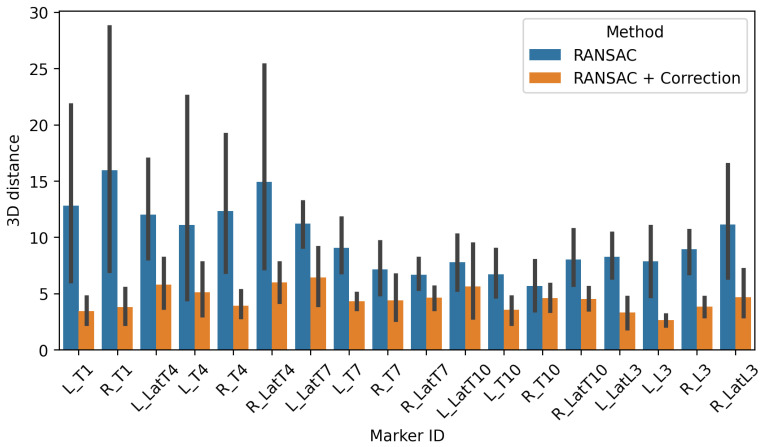
Distance error per marker from the resulting model to the radiograph positions. We compare a simple rigid registration of the skin to the radio-opaque markers using RANSAC (blue) with an additional step of pose correction using a kinematic model (orange).

**Figure 4 bioengineering-10-00874-f004:**
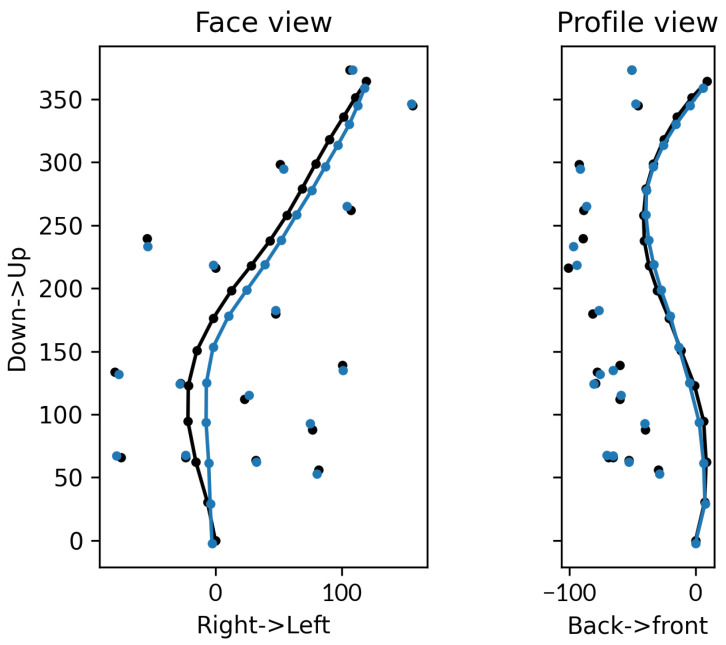
Predictions (blue) against the ground-truth (black) on the coronal and sagittal planes (scale in mm). The free dots represent the marker positions and the connected ones the vertebrae.

**Table 1 bioengineering-10-00874-t001:** Stiffness values at each joint of the model. Shear and compression are expressed in kN/m, flexion–extension, axial rotation and lateral bending in N.m/rad.

Joint Region	Shear	Compr.	Flex.-Ext.	Ax.-Rotation	Lat. Bending
Thoracic	262	1720	286	177	223
Lumbar	245	1720	143	498	149

**Table 2 bioengineering-10-00874-t002:** Accuracy (mean absolute error, MAE) of the avatar vertebrae in shape, positions and orientations.

ID	Positions (mm)				Orientations (deg)			Shape (mm)
	3D Distance	Anteropos.	Mediolat.	Inferosup.	Coronal	Sagittal	Axial	P.-t.-S. Distance
T01	1.32 (3.28)	1.05 (2.78)	0.7 (1.78)	0.11 (0.18)	1.67 (2.92)	1.98 (2.6)	3.58 (5.33)	0.31 (0.33)
T02	0.33 (0.7)	0.15 (0.26)	0.27 (0.66)	0.06 (0.08)	1.25 (1.23)	3.19 (2.34)	1.55 (1.09)	0.28 (0.27)
T03	0.17 (0.18)	0.14 (0.19)	0.06 (0.04)	0.05 (0.06)	1.33 (0.77)	7.22 (4.66)	1.87 (1.38)	0.28 (0.25)
T04	0.27 (0.28)	0.23 (0.26)	0.06 (0.06)	0.1 (0.12)	2.16 (1.44)	3.4 (3.69)	1.28 (0.86)	0.31 (0.28)
T05	0.18 (0.23)	0.14 (0.23)	0.04 (0.02)	0.07 (0.09)	2.77 (2.98)	1.52 (1.0)	1.67 (0.53)	0.29 (0.28)
T06	0.07 (0.06)	0.06 (0.06)	0.03 (0.02)	0.02 (0.01)	1.48 (1.55)	2.14 (1.47)	2.47 (2.23)	0.26 (0.24)
T07	0.24 (0.25)	0.15 (0.2)	0.09 (0.08)	0.15 (0.14)	1.04 (1.52)	2.23 (1.56)	1.17 (1.13)	0.31 (0.28)
T08	0.13 (0.06)	0.1 (0.06)	0.03 (0.02)	0.06 (0.04)	0.92 (0.79)	5.14 (2.36)	3.07 (3.09)	0.29 (0.31)
T09	0.15 (0.07)	0.12 (0.08)	0.06 (0.04)	0.03 (0.04)	1.17 (0.58)	4.35 (2.29)	2.29 (3.84)	0.28 (0.25)
T10	0.14 (0.06)	0.11 (0.08)	0.04 (0.03)	0.04 (0.02)	1.79 (0.96)	7.8 (3.5)	4.21 (3.44)	0.31 (0.3)
T11	0.21 (0.23)	0.16 (0.16)	0.05 (0.06)	0.12 (0.17)	1.4 (0.87)	8.11 (2.24)	2.63 (2.21)	0.3 (0.28)
T12	0.12 (0.15)	0.09 (0.14)	0.05 (0.07)	0.04 (0.04)	1.34 (0.63)	1.63 (1.56)	2.27 (2.02)	0.29 (0.27)
L01	0.19 (0.28)	0.14 (0.28)	0.07 (0.04)	0.06 (0.09)	1.78 (1.83)	3.84 (3.28)	1.79 (1.17)	0.31 (0.3)
L02	0.29 (0.27)	0.24 (0.27)	0.07 (0.06)	0.1 (0.12)	1.57 (1.61)	5.43 (4.81)	4.66 (8.08)	0.28 (0.26)
L03	0.31 (0.27)	0.28 (0.24)	0.05 (0.06)	0.1 (0.14)	2.62 (3.85)	4.74 (1.3)	2.79 (2.64)	0.28 (0.26)
L04	0.44 (0.23)	0.43 (0.22)	0.05 (0.06)	0.06 (0.06)	1.16 (1.22)	5.74 (2.15)	3.04 (5.16)	0.31 (0.29)
L05	0.61 (0.29)	0.58 (0.26)	0.1 (0.1)	0.15 (0.09)	1.65 (1.4)	2.37 (2.25)	3.7 (5.79)	0.3 (0.28)
All	0.31 (0.83)	0.25 (0.70)	0.11 (0.46)	0.08 (0.10)	1.59 (1.73)	4.17 (3.32)	2.59 (3.49)	0.29 (0.28)

**Table 3 bioengineering-10-00874-t003:** Accuracy of the predictions (MAE) of the vertebra positions and orientations according to the marker positions in lateral bending.

ID	Positions (mm)			Orientations (deg)		
	Anteropos.	Mediolat.	Inferosup.	Coronal	Sagittal	Axial
T01	4.53 (1.32)	2.38 (1.21)	3.62 (2.56)	6.17 (6.03)	2.3 (2.47)	2.2 (1.63)
T02	3.84 (1.9)	3.35 (3.84)	3.71 (2.47)	7.73 (6.57)	1.7 (1.36)	5.27 (4.24)
T03	3.47 (2.64)	4.98 (4.97)	2.6 (2.99)	6.39 (4.48)	1.17 (1.05)	3.41 (3.92)
T04	3.78 (3.13)	6.31 (5.98)	2.6 (1.71)	4.56 (4.52)	1.53 (0.98)	2.04 (1.04)
T05	4.21 (3.57)	7.13 (6.23)	1.86 (0.9)	4.87 (2.11)	1.69 (1.2)	2.96 (0.63)
T06	4.94 (4.25)	7.45 (6.25)	1.38 (1.17)	2.53 (1.45)	2.47 (1.99)	4.32 (2.0)
T07	5.55 (3.54)	7.12 (6.92)	0.92 (1.45)	2.03 (1.14)	2.35 (0.8)	4.4 (4.34)
T08	5.24 (2.64)	7.57 (6.86)	1.29 (1.03)	2.14 (2.15)	2.83 (1.35)	2.94 (1.33)
T09	4.39 (2.73)	9.0 (6.03)	1.49 (1.43)	3.65 (3.58)	1.8 (1.77)	2.99 (0.43)
T10	4.22 (3.44)	10.66 (2.45)	1.05 (0.95)	4.59 (6.29)	1.31 (1.07)	5.48 (4.8)
T11	3.32 (4.42)	12.01 (0.78)	1.53 (0.82)	4.19 (4.46)	1.55 (1.81)	7.28 (4.04)
T12	3.26 (2.65)	12.93 (1.76)	2.15 (0.37)	4.39 (4.06)	3.54 (1.08)	7.14 (4.02)
L01	5.42 (5.78)	14.42 (3.96)	1.36 (0.76)	7.69 (11.85)	8.1 (9.25)	8.01 (6.84)
L02	5.19 (5.13)	13.79 (4.57)	1.82 (1.19)	2.98 (2.99)	4.0 (1.19)	9.54 (6.36)
L03	5.39 (5.58)	11.09 (2.7)	0.94 (0.33)	2.9 (3.03)	1.23 (0.92)	7.1 (0.32)
L04	4.04 (3.97)	6.75 (3.64)	1.43 (0.19)	6.05 (7.55)	3.12 (2.57)	6.13 (5.55)
L05	3.78 (3.2)	7.38 (4.52)	1.59 (0.92)	4.76 (3.16)	2.95 (3.06)	13.98 (5.02)
All	4.39 (3.13)	8.49 (5.13)	1.84 (1.45)	4.57 (4.53)	2.57 (2.77)	5.60 (4.38)

## Data Availability

Not applicable.

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
