# Peer review of "Multi-Modal Data Correspondence for the 4D Analysis of the Spine with Adolescent Idiopathic Scoliosis"

_bioengineering, 2023, doi:10.3390/bioengineering10070874_

Round 1

Reviewer 1 Report

In the presented study the authors analyzed the use of kinematic predictions of motion in adolescent idiopathic scoliosis (AIS). Multi-modal data acquisition was created from the observation of external and internal mobility of the spine. The initial dataset consisted of 8 patients and the model was further validated on two subjects.

The authors were able to accurately reproduce the spinal mobility in AIS and create a more personalized approach for investigation of individual spinal mobility. The methods used are properly presented and the obtained results are promising.

The discussion section is rather short. I suggest the week points of the study, especially low number of subjects for dataset acquisition and validation should be mentioned. I would also recommend that possible future improvements of the described method are presented, as well as possible clinical implications in treatment of AIS.

Generally, the study is novel and its results will be important for future studies of the biomechanics in AIS. I propose the article to be accepted with minor revisions.

Author Response

Thank you for your comments on our submission on ”Multi-modal data correspondence for the 4D analysis of the spine with Adolescent Idiopathic Scoliosis”. We have added a note about the small number of patients in our study at the end of the discussion. This completes the passage dedicated to future improvements.

Reviewer 2 Report

Very interesting paper showing promising results in Adolescent Idiopathic Scoliosis management. We recommend accepting the paper in the current form and we encourage the authors to conduct a similar study on a higher number of patients, the only weak point of this study being the relatively low number of patients 

Minor English language corrections might be needed

Author Response

Thank you for your comments on our submission on "Multi-modal data correspondence for the 4D analysis of the spine with Adolescent Idiopathic Scoliosis".

Indeed, we found a few errors in grammar and spelling. We have corrected them:

  • "hormons" line 19 and 'groundtruth' line 131
  • The double "been" line 103 ;
  • Grammatical errors on lines 95, 183 and 242 and missing commas.

Reviewer 3 Report

1)  Line 10 makes it clear that the novelty of the work presented is the placement of markers for motion capture without the need for palpation, which is time-consuming and thus not suitable for young AIS patients as pointed out on line 43. It is not entirely clear how references 5-8 illustrate this point. Marking was performed by an experienced physiotherapist in the studies and no mention is made of time involved. With no information presented in the manuscript as to how much time is saved relative to palpation, a more logical motivation would be the utilization of less experienced medical personnel.   One way around the marker placement procedure would be to do away with them entirely as described in S Vafadar et al., A novel dataset and deep learning-based approach for marker-less motion capture during gait, Gait & Posture 86 (2021) 70-76.    2)  Line 60 describes marker placement as being in the neighborhood. This is somewhat vague with respect to general rules of motion capture marker positioning such as placement close to bone in order to minimize movement, and, specifically for the shoulder, placement where no movement is seen upon arm movement. More information is needed as to what instructions were given to the person placing the markers and the time required for a placement of a complete set.   3)  There may not be any explanation for the error trends shown in the tables, but some features draw attention. One is the much higher error for TO1 with respect to position compared to the others in Table 2 and sagittal having the greatest variation in orientation. Table 3 shows more uniformity but with deviations for mediolateral (mentioned on line 245) for T10-L03 and axial for L05.   4)  3D and 4D are mentioned numerous times in the manuscript, 2D only twice in connection with computations on lines 70 and 133. A comment could be added concerning if there is concern about error contribution from these. 

Author Response

Thank you for your relevant comments on our submission on ”Multi-modal data correspondence for the 4D analysis of the spine with Adolescent Idiopathic Scoliosis”. We have studied each of the points you have raised.   1) 5-8 illustrate the current state-of-the-art of the methods used to study spinal alignments with marker-based motion capture analysis. These elements of the literature do not comment on the palpation and placement times of the superficial markers. We have not calculated the time required to place these markers. This task is naturally simpler and quicker, since there is no need to palpate the vertebrae. We agree that this method can be applied by less qualified medical staff. Thus, we have added this comment to the discussion. The marker placement step for motion capture can be by passed in markerless acquisition methods. The reference you gave is very interesting and shows promising results in the 3D localization of joint centers. This study is essentially focused on the estimation of the 3D joints positions in gait analysis. However, our study focuses on bone movement in position and rotation with marker-based motion capture methods.   2) Placing mocap markers is a very delicate step, as the spinous processes of vertebrae are complicated to locate, particularly in scoliotic patients. Moreover, the position of the spinous process, the only palpable part of the vertebra, does not reflect the actual position of the bone. In scoliotic patients, it’s even more complicated, as the vertebra may be deformed. Our method uses a deformable model that changes its skin shape according to the positioning of its vertebrae. In the dynamic predictions part, we optimize the positions/rotations of the model’s vertebrae based on observations of the skin surface (i.e. markers). By fitting its marker positions, the model skin is deformed and predicts realistic vertebra positions and rotations with respect to its biomechanical properties. Precise positioning of the markers on the spinous processes is thus no longer an absolute requirement. For more details on the pose, the markers are positioned at different levels in order to cover the full thoracolumbar spine. Each marker close to the column is placed approximately 2 cm on either side of the spinous processes. The others (L_LatT4, R_LatT4, ...) are placed more laterally with respect to the width of the patient in order to cover the back surface.   3) You noted in Table 2 that T01 had a high positional error compared to the other vertebrae. This is due to a patient which has a particular T01 shape and on which the model could not be correctly registered. We have added a comment in the results section. The other errors Table 2 cannot be clearly explained. Orientation errors are given by computing an angle between the model and ground-truth vertebra orientation axis projected on a plane. This method is highly sensitive to the location of the anatomical landmarks used to draw the orientation axis (see ISB definitions Wu 2002). Ground-truth annotations are given by the SterEOS software and reconstruction ones by the model. We can note that the standard deviation of the errors in the sagittal plane is in the same magnitude in the axial plane. Table 3, the greater axial orientation error you noticed at L05 may be explained by the fact that we have no superficial marker at this level. As detailed in the discussion, the addition of markers in this area (pelvis for instance) would enable us to improve the accuracy of the model is this region. We added a comment in the results section.

4. This point is related to the effect of error in 2D localization of radio-opaque markers in the X-rays. As shown line 13, this error could have an effect on multi-modal matching and dynamic predictions. We can assume that the error in landmark location can be compensated by the fact that we have several markers arranged on the back. It would be interesting to see this in future experiments.